# Research Progress on Triarylmethyl Radical-Based High-Efficiency OLED

**DOI:** 10.3390/molecules27051632

**Published:** 2022-03-01

**Authors:** Jie Luo, Xiao-Fan Rong, Yu-Yuan Ye, Wen-Zhen Li, Xiao-Qiang Wang, Wenjing Wang

**Affiliations:** College of Chemistry and Chemical Engineering, Wuhan University of Science and Technology, Wuhan 430081, China; luojie@wust.edu.cn (J.L.); rongxiaofansnow@sina.com (X.-F.R.); yeyuyuan1998@hotmail.com (Y.-Y.Y.); lwz2688@163.com (W.-Z.L.)

**Keywords:** organic radicals, OLED, room-temperature luminescence, open-shell, doublet state

## Abstract

Perchlorotrityl radical (PTM), tris (2,4,6-trichlorophenyl) methyl radical (TTM), (3,5-dichloro-4-pyridyl) bis (2,4,6 trichlorophenyl) methyl radical (PyBTM), (N-carbazolyl) bis (2,4,6-trichlorophenyl) methyl radical (CzBTM), and their derivatives are stable organic radicals that exhibit light emissions at room temperature. Since these triarylmethyl radicals have an unpaired electron, their electron spins at the lowest excited state and ground state are both doublets, and the transition from the lowest excited state to the ground state does not pose the problem of a spin-forbidden reaction. When used as OLED layers, these triarylmethyl radicals exhibit unique light-emitting properties, which can increase the theoretical upper limit of the OLED’s internal quantum efficiency (IQE) to 100%. In recent years, research on the luminescent properties of triarylmethyl radicals has attracted increasing attention. In this review, recent developments in these triarylmethyl radicals and their derivatives in OLED devices are introduced.

## 1. Introduction

As important intermediates in chemical reactions [1,2,3] and human metabolism [4,5,6], organic radicals have attracted great attention in the research of chemical reaction mechanisms [7,8,9], polymerization [10,11], catalysis [12,13], and biochemistry [14,15]. Due to their unique unpaired electron structure [16], stable organic radicals possess fascinating optical, electronic, and magnetic properties. Additionally, as a result, they have been widely used in the fields of polarization optics [17,18,19,20], organic optoelectronics [21,22,23,24], spintronics [25,26,27,28], and molecular magnetism [29,30,31,32]. However, in most cases, organic radicals have difficulty remaining stable under ambient conditions. In 1900, Gomberg et al. found the first relatively stable organic radical, trityl radical (TPM) [33]. TPM could remain stable in some solvents, but quickly polymerized in the air [34]. It was not until 1970 that Ballester et al. synthesized perchlorotrityl radicals (PTM) [35], which could remain stable at room temperature.

Organic light-emitting diodes (OLEDs) [36,37,38] are light-emitting devices that can convert electrical energy into light. In 1963, Pope et al. discovered that anthracene could emit light under an electric field [39], and the first OLED device came about by using organic materials as the light-emitting layer in 1987 [40]. Compared with other light-emitting devices, OLEDs have the advantages of light weight, low energy consumption, high contrast, fast response, and the ability to produce transparent devices [41,42,43,44,45]. In 2015, Li’s group proposed the doublet theory and presented the first OLED device produced with organic radicals [46]. Different from traditional phosphorescence [47,48,49], thermally activated delayed fluorescence (TADF) [50,51,52], triplet–triplet annihilation (TTA) [53,54,55], and localized and charge-transfer hybrid states (HLCTs) [56,57,58], the proposal of this special doublet luminescence mechanism increased the theoretical upper limit of OLED’s internal quantum efficiency (IQE) to 100%, while traditional fluorescent materials were plagued by the poor utilization of triplet excitons. This has made scientists’ research of organic radicals in the field of OLEDs extremely attractive. Recently, the development of chiral molecules exhibiting circularly polarized luminescence (CPL) has received extensive attention. In circularly polarized OLED (CP-OLED), due to its circular structure, the power is reduced by depolarization during photoelectric conversion [59]. At the same time, the problem of singlet and triplet exciton collection is well resolved using chiral phosphorescent materials, such as iridium(III) complexes or thermally activated delayed fluorescence (CP-TADF) emitters, which improve the performance of OLED [60]. Due to its unique electronic structure, triarylmethyl radical can achieve controllable CPL luminescence properties by applying a longitudinal magnetic field, supramolecular chiral co-assembly and inducing chiral liquid crystal, which shows its high potential for application in the field of CP-OLED [61]. In this work, we summarize recent progress in different triarylmethyl radicals that have been applied in the OLED field, including perchlorotrityl radicals (PTM), tris (2,4,6-trichlorophenyl) methyl radicals (TTM), (3,5-dichloro-4-pyridyl) bis (2,4,6 trichlorophenyl) methyl radicals (PyBTM), (N-carbazolyl) bis (2,4,6-trichlorophenyl) methyl radicals (CzBTM), and their derivatives.

## 2. Introduction to OLED

### 2.1. OLED’s Working Principle and Material Structure Composition

As shown in Figure 1a, OLED devices are composed of organic thin-film layers stacked between a cathode and an anode, including an electron injection layer (EIL), electron transport layer (ETL), light-emitting layer (EML), hole transport layer (HTL), and hole injection layer (HIL) [62]. The whole process of OLED electroluminescence is shown in Figure 1b. Under an external electric field, electrons and holes are injected from the cathode and anode into the EIL and HIL, respectively, and then these electrons and holes pass through ETL and HTL to reach EML. Finally, the electrons and holes combine to generate excitons in EML under the action of coulomb static electricity [63,64]. As the excitons are excited, they need to release energy in some way and return to the ground state [65,66]. There are two main ways to release energy: one is in the form of light radiation, and the other is in the form of non-radiation [67]. Generally, the radiation and non-radiation processes are similar to ebb and flow, which are related to the photoelectric conversion rate of the devices [68,69,70]. Only when as many excitons as possible are inactivated in the form of light, the efficiency of the device will be high. Therefore, the light-emitting layer plays a very important role in OLED’s performance and light color [71,72].

The external quantum efficiency (EQE) is the most direct parameter to describe the efficiency of an OLED device. It is defined as the ratio of the number of photons emitted from the device to the number of electrons injected into the device under an electric field. Its mathematical expression is as follows:

Φ_EQE_ = Φ_IQE_ × η_out_ = χ × Φ_PL_ × η_r_ × η_out_(1)

In the formula, Φ_EQE_ represents the external quantum efficiency of an OLED device; Φ_IQE_ represents the internal quantum efficiency of the device, which is the efficiency of injecting electrons to produce photons in the light-emitting layer; η_out_ represents the efficiency of the optical coupling-out of light; χ represents the exciton utilization rate, which is the light conversion rate of excitons in the light-emitting layer; Φ_PL_ represents the absolute photoluminescence quantum efficiency of the light-emitting layer in the solid film; η_r_ represents the recombination ratio of carriers (electrons and holes) in the OLED device [73,74]. Therefore, Φ_EQE_ is not only proportional to Φ_IQE_, but also positively related to the coupling-out efficiency of light (η_out_), and its magnitude is usually between 20% and 30% [75]. As a result, scientists have been committed to researching various luminescent materials to improve the related performance.

### 2.2. Classification of OLED Lighting Methods

According to the relevant principles of quantum spin statistics, when the OLED light-emitting layer, composed of closed-shell fluorescent molecules, is electrically excited, holes and electrons are injected into the anode and cathode of the devices. Then, the holes and electrons are transported to the light-emitting layer and combine to generate two excitons. The generation ratio of singlet and triplet excitons is 1:3, which means that only 25% of the excitons are in the singlet state, and 75% of the excitons are in the triplet state [76]. However, according to Pauli’s exclusion principle, it is spin forbidden for triplet excitons to return to the ground state, and only the singlet excitons can release energy in the form of emitted photons and return to the ground state [77]. Therefore, the theoretical effective exciton generation ratio of traditional fluorescent OLEDs and the corresponding upper limit of IQE are only 25%. Considering the coupling-out light efficiency is 20%–30% [75], the maximum EQE of devices is only 5–7.5%.

To improve the utilization of triplet excitons in fluorescent OLEDs, some effective mechanisms and methods have been proposed in recent years, such as phosphorescence, thermally induced delayed fluorescence (TADF), triplet–triplet quenching (TTA), and localized and charge transfer hybrid state (HLCT) (Figure 2) [62]. For example, in phosphorescent materials, 25% of singlet excitons can be converted into triplet excitons through inter-system transition, and OLEDs prepared from phosphorescent materials can theoretically achieve 100% IQE. However, there are still problems to be dealt with, such as the high cost of the use of precious metals, and the potential environmental pollution problems [78,79]. Although TADF [80], TTA [81], and HLCT [82] can improve the efficiency of OLED devices, there are more challenges of molecular design or material synthesis to use triplet excitons effectively.

In 2015, Li’s group creatively proposed the theory of dual-state light-emitting OLED to avoid the problems of singlet and triplet excitons (Figure 3) [46]. Different from closed-shell organic fluorescent molecules, when the open-shell organic radicals are electrically excited, the ground state and the excited state of the radicals are both doublet states. According to Pauli’s exclusion principle, the transition of organic radical electrons from the lowest excited state back to the ground state is completely spin allowed. Thus, OLEDs based on light-emitting organic radicals can achieve 100% IQE theoretically. However, since organic radicals are special organic compounds in metastable states, which have less chemical bonds than normal compounds and contain one unbonded single electron, organic radicals have high chemical activity and are often used as reaction intermediates under normal circumstances. Thus, they will quickly transform into dimers or react with oxygen to form closed-shell molecules [17], which make it difficult to collect and use [83].

Reasonable molecular design can improve the stability of single-electron organic radicals. In 1900, Moses Gomberg synthesized the first “stable” organic radical, trityl radical (TPM) [16]. TPM was partly stable since it was stable in organic solvents but would easily react with oxygen to form dimers after removing the solvents (Figure 4). This was a major discovery in the field of stable organic radicals, and it also paved the way for the study of stable organic radicals.

In recent years, an increasing number of stable organic radicals have been discovered [84]. According to the atomic center, where the single electron is located, organic radicals can be divided into three categories: carbon-based organic radicals, nitrogen/oxygen-based organic radicals, and organic radicals based on other atoms. In these radicals, the conjugation effect [85] and protection of large steric groups [86] are often used to improve the stability of organic radicals. However, only methyl radicals with carbon as the core [87] can remain stable and exhibit light emission at room temperature. These methyl radicals can be divided into four categories: PTM, TTM, PyBTM, and CzBTM.

In this review, triarylmethyl radicals and their derivatives are described in detail, and their applications in OLED device-related materials are explained. The photoluminescence quantum efficiency (PLQE) of different tritylmethyl radicals is summarized in Table 1. We hope that we can not only provide more research ideas on organic radicals, but also provide useful information about the design and preparation of OLED devices.

## 3. PTM-Based Luminescent Organic Radicals

In 1970, Ballester et al. synthesized perchlorotrityl radical (PTM), which was the first stable triarylmethyl radical to achieve room-temperature luminescence, and it could exist in the environment for decades [107]. The unique stability of PTM came from the shielding effect of the ortho-position chlorine atoms of the three benzene rings on the central carbon, and the steric hindrance effect limited the formation of any form of dimers of PTM [108]. At room temperature, PTM could exhibit a faint orange-red light emission at 605 nm in a non-polar solvent, but the fluorescence quantum efficiency (QE) of PTM was very low. In cyclohexane, its photoluminescence quantum efficiency (PLQE) was only 0.015, and it would reduce two chlorines under light irradiation (Figure 5) [109].

Due to its low fluorescence quantum yield, PTM was not as attractive until Lambert et al. used the Suzuki reaction to replace one para-position Cl of the benzene rings of PTM with Br (Figure 6a) in 2004 [110]. Later, they introduced a triphenylamine structure on PTM, and modified different electron-withdrawing groups on the triphenylamine part to obtain a series of luminous PTM derivatives (Figure 6b) [111,112]. Their emission wavelengths were between 600 nm and 900 nm. Additionally, their PLQE showed an increasing trend with a shorter wavelength, with a maximum of 0.37, which was nearly 25 times higher than that of unmodified PTM, but it was still much lower in polar solvents. These PTM derivatives significantly reduced the electron transfer rate of organic radicals (Figure 6c), which improved the fluorescence efficiency of organic radicals. Additionally, it was also proposed that donor-acceptor (D-A)-type PTMs were mixed-valence compounds (Figure 6d) [111], and the strong electron hybridization of the donor-acceptor could change its luminescence characteristics, local excitation (LE) state, and charge transfer (CT) state (Figure 6e) [113]. On this basis, in 2013, Veciana et al. connected a tetrathiafulvalene (TTF) unit to a PTM unit and obtained a stable dimer radical of the binary group (Figure 6f) [114].

In 2018, Li’s group combined triphenylamine (TPA) and different site-substituted carbazoles (PCz and 3PCz) with PTM to synthesize three different PTM-like radicals, PTM-PCz, PTM-3PCz, and PTM-TPA (Figure 7a) [88]. It was found that the introduction of electron-donating triphenylamine (TPA) and carbazole derivatives (PCz and 3PCz) at different positions could not only significantly increase the PLQE of PTM-like radicals in non-polar solutions, but also greatly improved the light stability of these radicals. As shown in Figure 7b, compared with PTM, the PLQE of these radicals in cyclohexane increased 44 times, 57 times, and 26 times, respectively, and the half-life increased 51 times, 185 times, and 6808 times, respectively. At the same time, their UV–vis absorptions and fluorescence spectra showed redshifts to different degrees compared with PTM (absorption peak shifted from 566 nm to 680 nm, and emission peak shifted from 604 nm to 767 nm) (Figure 7c). It was also found that different solvents had different influence trends on their fluorescence spectra. As shown in Figure 7d, the fluorescence emission of substituted PTM increased with the increase in solvent polarity, accompanied by an obvious redshift, while the fluorescence emission of unsubstituted PTM had little effect on solvents of different polarities. Additionally, the PLQE of substituted PTM decreased sharply as the polarity of the solvent increased, due to the charge transfer phenomenon between the electron donor of the carbazole derivative/triphenylamine and the electron acceptor of the PTM unit.

Later in 2019, Li’s group introduced 9-(naphthalene-2-yl)-9H-carbazole (NCz) and 1,3-di(9Hcarbazol-9-yl) benzene (PDCz) into PTM, and obtained PTM-3NCz and PTM-PDCz with deep red/near infrared (NIR) emission peaks at about 700 nm (Figure 8a) [89]. Their photoluminescence quantum yield (PLQY) reached 54% and 15%, they exhibited strong photochemical stability, and their half-life could reach several months under pulsed ultraviolet laser irradiation. The OLED device obtained by treating the emitter with PTM-3NCz via spin coating had a maximum EQE of 5.3%, which was very high for pure organic deep-red/near-infrared (NIR) emitters (Figure 8b,c).

In 2020, Perepichka et al. synthesized a white crystal of tris(iodoperchlorophenyl) methane (3I-PTM^H^) with a radical concentration of 4%. As shown in Figure 9a,b, the white crystal emitted red light under the off-white solid light at room temperature, and its PLQY was as high as 91%. Additionally, the immobilized iodinated radical showed excellent light stability (half-life of more than one year) (Figure 9c) and a relatively long luminescence lifetime (69 ns) (Figure 9d) [90].

## 4. TTM-Based Luminescent Organic Radicals

In 1987, Armet et al. synthesized a triarylmethyl radical with fewer chlorine atoms—TTM (Figure 10) [115]. Compared with PTM-based triarylmethyl radicals, TTM-based triarylmethyl radicals had fewer halogen atoms on the benzene ring, which made their steric hindrance smaller, and the modification simpler. Additionally, their light-emitting color was slightly blue-shifted compared with PTM (orange-yellow at 562 nm). However, like PTM, the PLQE of TTM was also very low (only 0.008 in cyclohexane solution), and it was unstable under light irradiation [91].

In 1994, Julia et al. reported a series of breakthrough research results on the synthesis of TTM derivatives, which effectively adjusted the luminescence properties of TTM-based triarylmethyl radicals [116,117,118]. In 2006, they connected the electron-donating carbazole group to TTM through carbon–nitrogen coupling and successfully obtained a triarylmethyl radical (TTM-1Cz) emitting strong red light (Figure 11a) [119]. The PLQE of the radical in cyclohexane reached 53%, which was approximately 18 times higher than that of substituted TTM. Later, they introduced a variety of carbazole derivatives and indole groups, and obtained a series of TTM derivatives with higher PLQE (Figure 11b).

It is worth noting that in 2015, Li’s group used TTM-1Cz-doped film as the light-emitting layer, and successfully fabricated a deep red OLED device based on triarylmethyl radicals for the first time. The EQE of the device reached 2.4%, achieving a breakthrough on a new path to the 25% upper limit of OLED’s IQE (Figure 12a) [46]. Later, they continued to optimize the structure of this OLED device, changed the proportion of triarylmethyl radicals, and increased the EQE of the device to 4.3% (Figure 12b) [92]. It was also found that when TTM-1Cz was doped into its precursor HTTM-1Cz at a ratio of 14%, the ratio of doublet excitons in the device was close to 100%. On this basis, they replaced the carbazole group in TTM-1Cz with several weak electron-donating benzimidazole groups, and obtained an orange-emitting organic radical. Its PLQE had increased dozens of times compared with the unsubstituted TTM. Additionally, it was then applied to obtain orange OLED with EQE as high as 5.4%, which expanded the light color of OLED based on triarylmethyl radicals (Figure 12c) [93].

Then, in 2018, Li’s group selected carbazoles with different N positions to modify TTM and obtained four stable triarylmethyl radicals, αPyID-TTM, βPyID-TTM, γPyID-TTM, and δPyID-TTM [94], which exhibited red light emissions (Figure 13). Different from most organic radicals, these radicals have extremely high luminous efficiency in different polar solvents. For example, in chloroform, the luminous efficiency is 91%, 89%, 32%, and 99%, respectively, which is 12~38 times higher than TTM (PLQE = 2.6%) and 6~20 times higher than TTM-1Cz (PLQE = 5%). The OLED devices obtained by these organic radicals all had high EQE, and the βPyID-TTM device had a maximum EQE of 12.2%, which was a relatively high level in pure red OLED devices, due to the change in the donor-acceptor structural motif [120]. At the same time, they synthesized two disubstituted radicals, biscarbazoline-substituted TTM, 2αPyID-TTM and 2δPyID-TTM, which also realized excellent luminescence properties and better thermal and electrochemical stability. Additionally, because 2αPyID-TTM and 2δPyID-TTM had more protonation sites, they exhibited better proton response characteristics than single-substituted radicals [95].

In the same year, Li’s group modified the carbazole group in TTM-1Cz to 4-azacarbazole and prepared a red OLED with an EQE of 10.6% (Figure 14a) [121]. They also modified the polycarbazole substituents at different substitution positions and the strong electron-donating group triphenylamine (TPA) on TTM to obtain the infrared-emitting radical TTM-PCz (cyclohexane 663 nm), TTM-3PCz (cyclohexane 664 nm), and TTM-TPA (cyclohexane 728 nm) (Figure 14b) [88]. The PLQEs of these radicals in cyclohexane were 0.04, 0.29, and 0.23, respectively. Additionally, all the three radicals had super light stability, which could be comparable to closed-shell molecules. Additionally, Duan et al. prepared an OLED-conductive film by doping 10% TTM-1Cz into HTA-CN, which increased the conductivity hundreds of times (Figure 14c) [122].

In 2019, Li’s group modified a polystyrene backbone with CzBTM and obtained a light-emitting radical polymer, PS-CzTTM, which was the first synthesis of light-emitting triarylmethyl polymers (Figure 15a) [123]. PS-CzTTM had paramagnetism and good thermal properties. Additionally, the solid state could exhibit bright deep red light emissions when exposed to ultraviolet light at room temperature. The films prepared by PS-CzTTM in cyclohexane solution or via spin coating had high luminescence quantum yields, with PLQEs of 37.5% and 24.4%, respectively. Additionally, its light stability was 300 times higher than that of TTM (Figure 15b). In the same year, they modified 9-(naphthalene-2-yl)-9H-carbazole (NCz) and phenyl-phenothiazine (PPTA) onto TTM and obtained TTM-3NCz and TTM-PPTA with high stability [89], whose PLQEs were 0.29 and 0.02, respectively, and the PLQE of TTM-3NCz was 36 times higher than that of TTM.

In 2021, Kuehne et al. used a special synthesis method to partially or fully bromine the chlorine in TTM and synthesized new triarylmethyl radicals, TBr_3_Cl_6_M, TBr_6_Cl_3_M, and TTBrM (Figure 16) [96]. They provided new ideas for the study of the properties of TTM-based organic radicals and the design of more complex triarylmethyl open-shell molecules, which was conductive to the development of small organic open-shell molecules in the field of optoelectronics.

In 2021, Zhou et al. replaced the carbazole in TTM-Cz with diphenylamine (DPA), dibenzidine (DBPA), and difluorenamine (DFA) to obtain a series of new diarylamines substituted TTM derivatives, TTM-DPA, TTM-DBPA, and TTM-DFA [97]. The maximum photoluminescence wavelengths of these triarylmethyl radicals were 705 nm, 748 nm, and 809 nm, due to the introduction of stronger electron-donating groups. Additionally, their PLQEs were 65%, 28%, and 5%, while their half-lives were 15 times, 45 times, and 73 times higher than those of TTM-Cz. These works have greatly enriched the development of stable TTM-based triarylmethyl radicals and provided more new ideas for their application research in the OLED field.

## 5. PyBTM-Based Luminescent Organic Radicals

In 2014, Nishihara et al. replaced a trichlorobenzene in TTM with dihydropyridine, and successfully synthesized an orange-red light radical, PyBTM, whose light stability was 115 times higher than that of TTM. Additionally, its PLQE was as high as 0.81 in the EPA solvent at 77 K (Figure 17a) [98]. Later, they introduced Au into PyBTM and synthesized the first metal luminescent radicals (Figure 17b). Compared with PyBTM, its fluorescence wavelength, quantum yield, and light stability had been further improved [124]. In addition, they replaced the two chlorine atoms on the pyridine ring in PyBTM with other halogen atoms to obtain X_2_PyBTM (X = F, Cl, Br) and observed the changes in the photophysical properties. They found that the luminous efficiency gradually decreased, but stability increased in turn (Figure 17c) [99].

In 2018, Nishihara et al. doped PyBTM into H-PyBTM crystals at different concentrations and obtained a fluorescence quantum yield of 0.89 at room temperature (Figure 18a). They found that when the concentration of PyBTM was 0.10, the emission spectra of the H-PyBTM crystal could change with the change in the applied magnetic field at extremely low temperatures (4.2 K) (Figure 18b) [100]. In the same year, they replaced more benzene rings with pyridine rings based on PyBTM, generating new organic radical, bisPyBTM (Figure 18c), and observed the solid-state luminescence of the radical at 77 K (Figure 18e) [101]. Additionally, the light stability of the radical had been further improved, and its half-life was 42 times that of PyBTM and 3000 times that of TTM (Figure 18d).

In 2021, Uchida et al. replaced chlorine on the pyridine ring of PyBTM with fluorine and substituted bromine and aromatic rings at the three positions of the two benzene rings, resulting in a shorter emission wavelength organic radical with low photostability and higher fluorescence quantum yield. Then, they synthesized six new stable organic radicals (e.g., *m*Br_2_-F_2_PyBTM, *m*Ph_2_-F_2_PyBTM, *m*ClPh_2_-F_2_PyBTM, *m*Py_2_-F_2_PyBTM), whose light stability had been further strengthened (Figure 19) [102].

In 2021, Kusamoto et al. replaced the three benzene rings in TTM with pyridine rings and synthesized a new type of light-emitting organic radical, trisPyM (Figure 20a) [103]. The new organic radical had strong photostability in organic media, and its half-life in CH_2_Cl_2_ was 4 times, 160 times, and 10,000 times higher than that of bisPyTM, PyBTM, and TTM, respectively (Figure 20b). As there were more pyridyl moieties in the TTM skeleton, the photostability of the radicals increased. In addition, a method of using N and metal to form coordination polymers was proposed to prepare a two-dimensional (2D) honeycomb spin-lattice, which provided a new idea for the preparation of polymer light-emitting materials (Figure 20c). Although the photostability of trisPyM was significantly improved compared with PTM and TTM, its PLQE level was not satisfactory. In the same year, they changed the position of the nitrogen atom through reasonable molecular design and synthesized a new stable organic radical, metaPyBTM [104], whose nitrogen atom was from para to meta. Although its PLQE and light stability were equivalent to TTM, the radical still showed obvious solid photoluminescence even at room temperature, which provided a new opportunity for organic radicals in the field of optical functions.

## 6. CzBTM-Based Luminescent Organic Radicals

In 2018, Li’s group successfully prepared a new type of stable organic radical, CzBTM, at room temperature (Figure 21a) [105], by replacing a benzene ring in TTM with a carbazole ring. CzBTM exhibited a deep red light emission of about 697 nm in a cyclohexane solution, and its PLQE was 0.02. Its light stability and thermal stability were very high. For example, the light stability was 273 times higher than that of TTM (Figure 21b). The CzBTM-based OLED exhibited a deep red light emission, and the EQE of the device was 0.66%. The successful synthesis of CzBTM provided a new method for the study of luminescent radical systems.

In the same year, Li’s group replaced the carbazole substituent with the pyridoindole substituent of the N heterocyclic ring based on CzBTM to prepare benzhydryl radical derivatives, PyID-BTM, which had higher luminous efficiency (Figure 22a) [106]. The photoluminescence efficiency of PyID-BTM was 10 times higher than that of carbazole, and the PLQE in cyclohexane reached 19.5%. On this basis, deep red light-emitting OLED devices were prepared, and the EQE of the device reached 2.8% (Figure 22b).

## 7. Conclusions

In conclusion, recent progress in room-temperature light-emitting triarylmethyl radicals, such as PTM, TTM, PyBTM, CzBTM, and their derivatives, applied to OLED is summarized. Due to the unique doublet emission mechanism, the key issue of harvesting the triplet energy in an OLED is circumvented. The radiative decay of the doublet is totally spin-allowed, so that the light-emitting triarylmethyl radicals can increase the upper limit of OLED’s IQE to 100%. From the current point of view, there is still a long way to go in the research on light-emitting triarylmethyl radicals. However, limited room-temperature light-emitting triarylmethyl radicals are reported; the research in this field is still in its infancy. Better light-emitting triarylmethyl radicals and more effective device structures are highly desirable to expand the width and depth of the light-emitting mechanism. In addition, triarylmethyl radicals with high stability, feasible solubility, better emissions in a solid state, large stock shifts, and longer emitting wavelengths are also highly desirable for their applications in OLED devices and various fields. It is anticipated that investigation on stable triarylmethyl radicals will greatly promote the development of OLED and can also be applied in various fields with unique advantages.

## Figures and Tables

**Figure 1 molecules-27-01632-f001:**
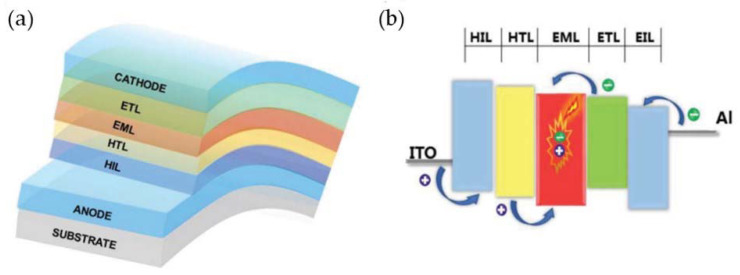
(**a**) The structure and (**b**) working principle of OLED devices. Reprinted with permission from ref. [62]. Copyright 2018, Copyright Advanced Optical Materials.

**Figure 2 molecules-27-01632-f002:**
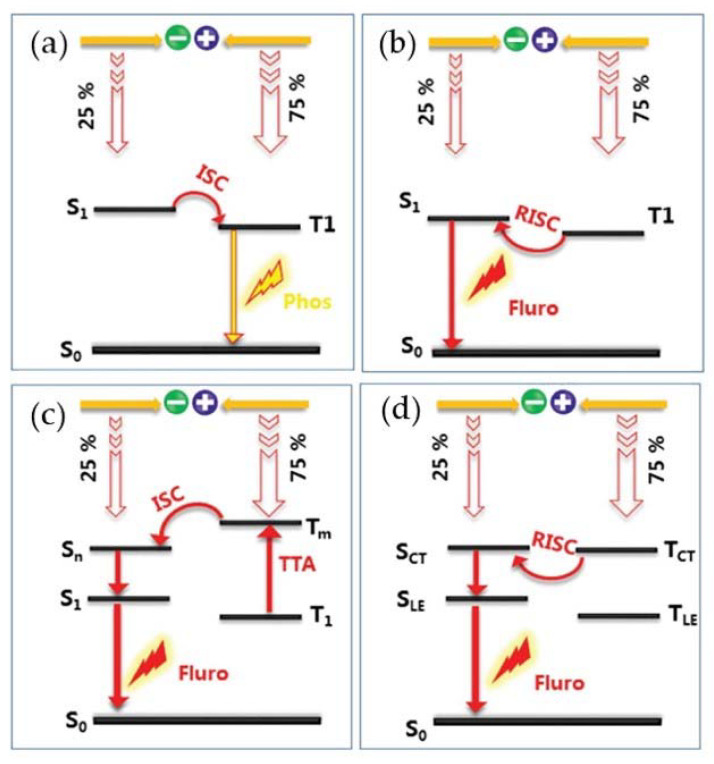
Mechanisms of different photophysical processes: (**a**) fluorescence, (**b**) TADF, (**c**) TTA, (**d**) HLCT. Reprinted with permission from ref. [62]. Copyright 2018, Copyright Advanced Optical Materials.

**Figure 3 molecules-27-01632-f003:**
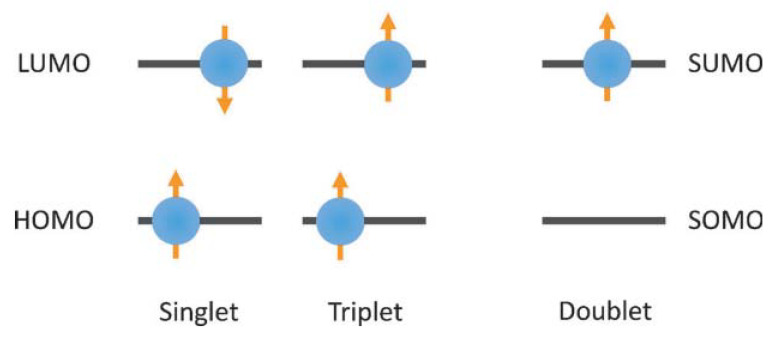
Excited state spin configurations of closed shell and double state open shell molecules. Reprinted with permission from ref. [46]. Copyright 2015, Copyright Angewandte Chemie International Edition.

**Figure 4 molecules-27-01632-f004:**
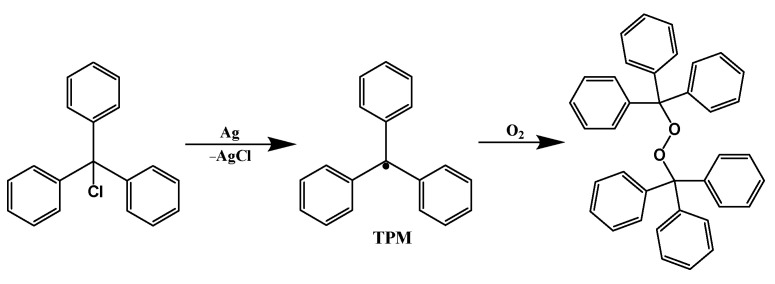
Preparation of TPM and their oxidation process.

**Figure 5 molecules-27-01632-f005:**
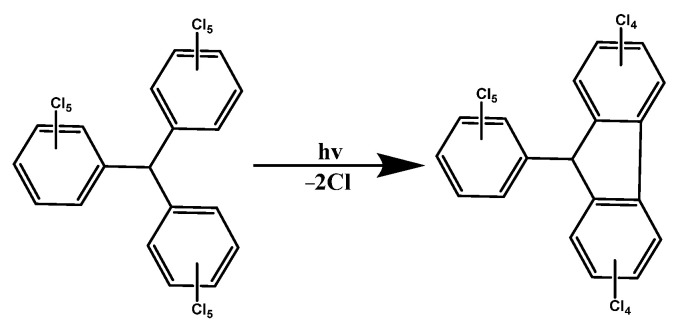
PTM formed rings under light irradiation.

**Figure 6 molecules-27-01632-f006:**
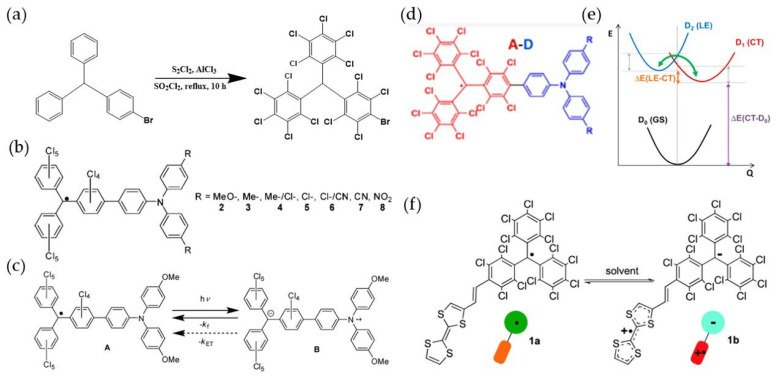
(**a**) Synthesis of Bromo-PTM. (**b**) Structure of PTM-TPA and its derivatives. (**c**) Fluorescence emission process of PTM derivatives. Reprinted with permission from ref. [111]. Copyright 2009, Copyright The Journal of Physical Chemistry C. (**d**) Conceptual diagram of proposed donor-acceptor type PTM. (**e**) Schematic diagram of the potential energy surfaces of the ground state (GS), CT state, and LE state of the radical. Reprinted with permission from ref. [113]. Copyright 2020, Copyright Journal of the American Chemical Society. (**f**) The intramolecular electron transfer process between TTF unit and PTM unit. Reprinted with permission from ref. [114]. Copyright 2013, Copyright Journal of the American Chemical Society.

**Figure 7 molecules-27-01632-f007:**
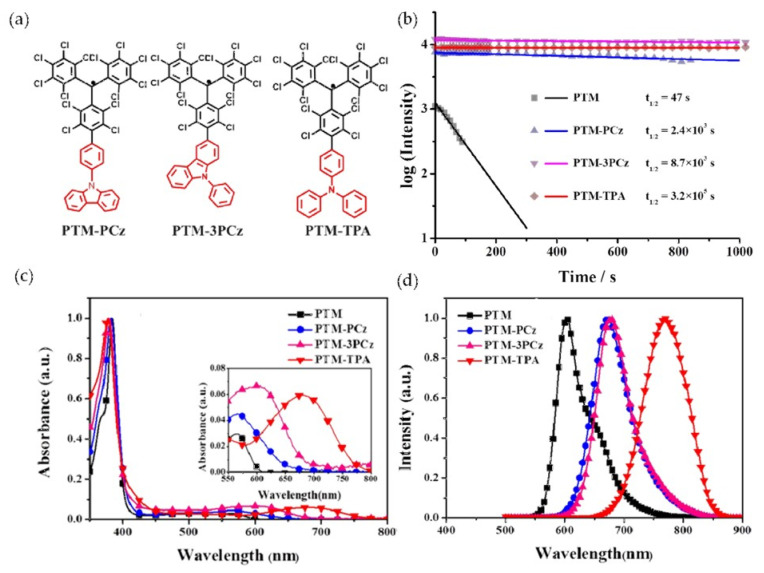
PTM derivatives: (**a**) molecular structures, (**b**) half-life, (**c**) UV–vis absorptions, (**d**) fluorescence emissions (excitation at 380 nm). Reprinted with permission from ref. [88]. Copyright 2018, Copyright Physical Chemistry Chemical Physics.

**Figure 8 molecules-27-01632-f008:**
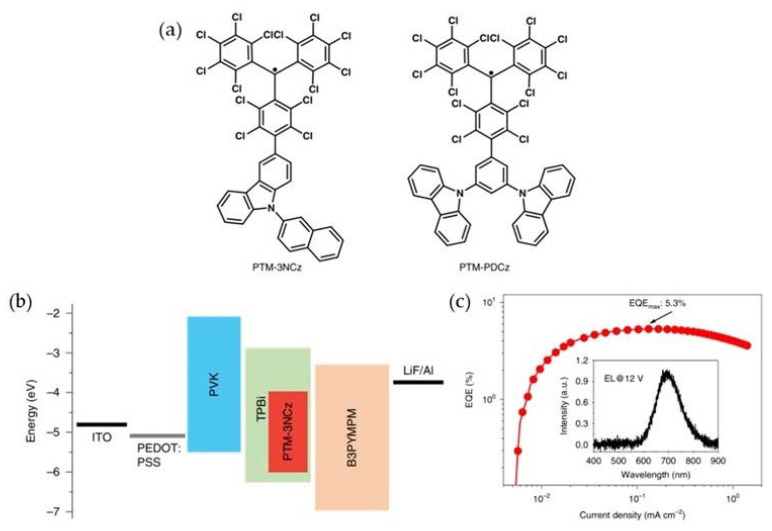
(**a**) Chemical structures of PTM-3NCz and PTM-PDCz. (**b**) Schematic structure of PTM-3NCz-based OLEDs. (**c**) EQE of the OLED device as a function of current density; inset: the full electroluminescence spectrum at 12 V. Reprinted with permission from ref. [89]. Copyright 2019, Copyright Nature Materials.

**Figure 9 molecules-27-01632-f009:**
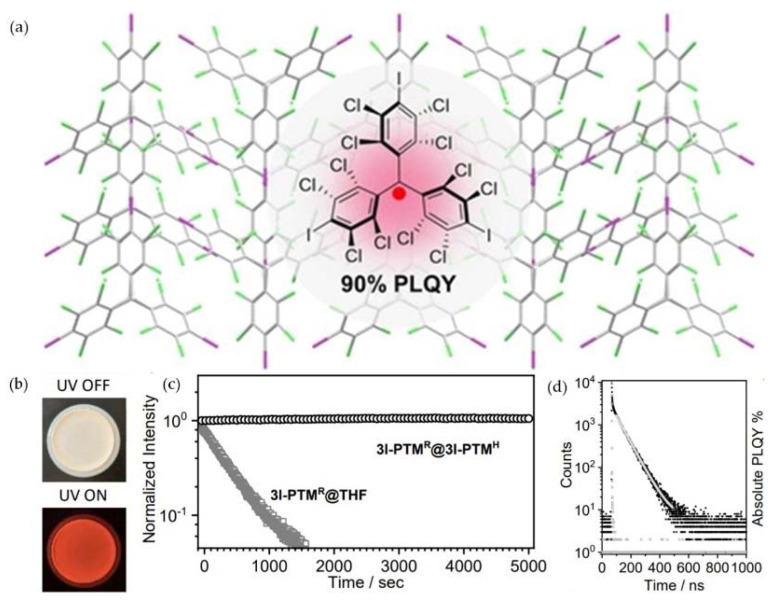
(**a**) The high PLQY of 3I-PTM^H^ doped with 4% 3I-PTM^R^. (**b**) Photographs of 3I-PTM^R^@3I-PTM^H^ before and after UV irradiation. (**c**) Half-life and (**d**) luminescence lifetime of 3I-PTM^R^@3I-PTM^H^. Reprinted with permission from ref. [90]. Copyright 2020, Copyright Angewandte Chemie International Edition.

**Figure 10 molecules-27-01632-f010:**
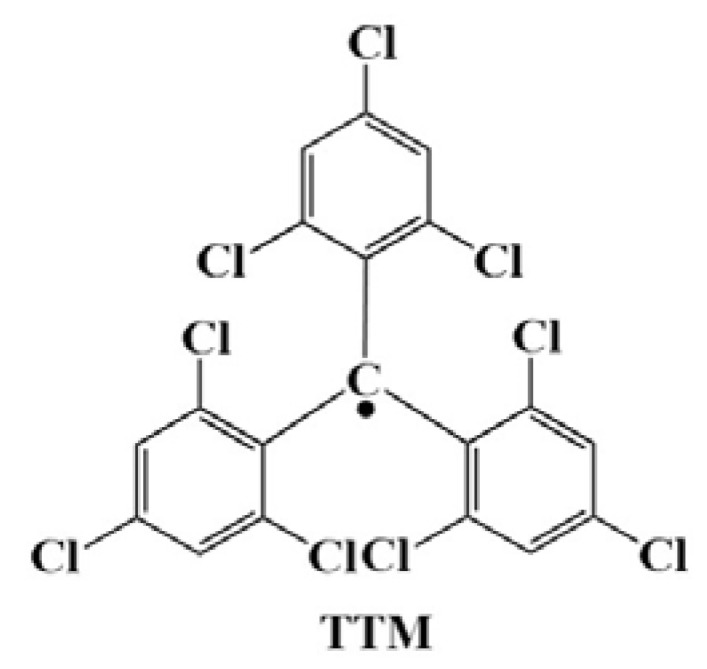
Structure of TTM.

**Figure 11 molecules-27-01632-f011:**
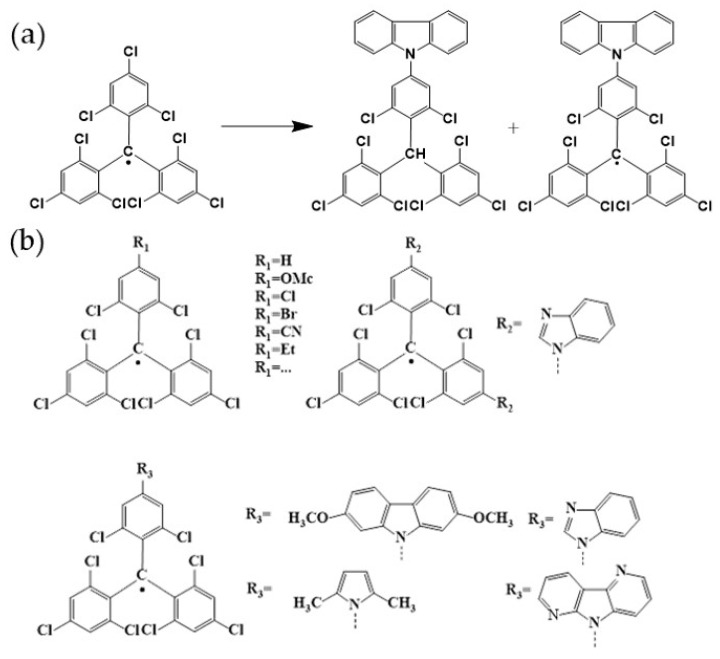
(**a**) TTM conversion to TTM-1Cz. (**b**) Molecular structures of carbazole and indole derivatives of TTM.

**Figure 12 molecules-27-01632-f012:**
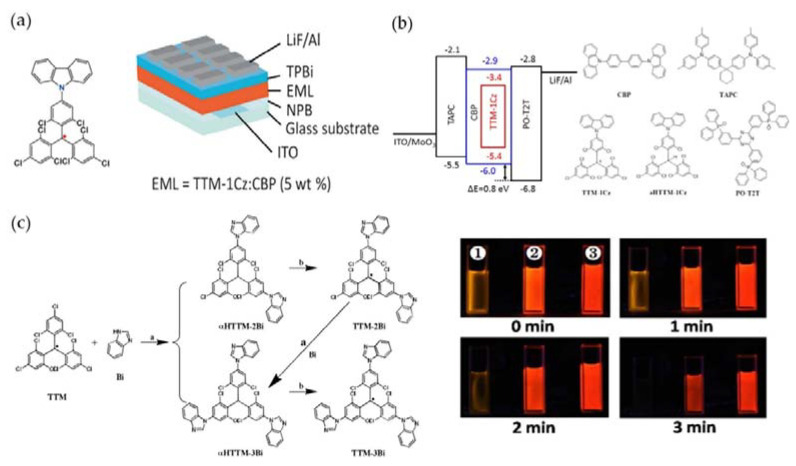
(**a**) 5% TTM-1Cz-doped OLED device. Reprinted with permission from ref. [46]. Copyright 2015, Copyright Angewandte Chemie International Edition. (**b**) The optimized OLED energy level diagram and its material molecular structure. Reprinted with permission from ref. [92]. Copyright 2016, Copyright ACS Applied Materials & Interfaces. (**c**) Synthesis of TTM-2Bi and TTM-3Bi, and photochemical stability of 1: TTM; 2: TTM-2Bi; 3: TTM-3Bi. Reprinted with permission from ref. [93]. Copyright 2017, Copyright Chemistry of Materials.

**Figure 13 molecules-27-01632-f013:**
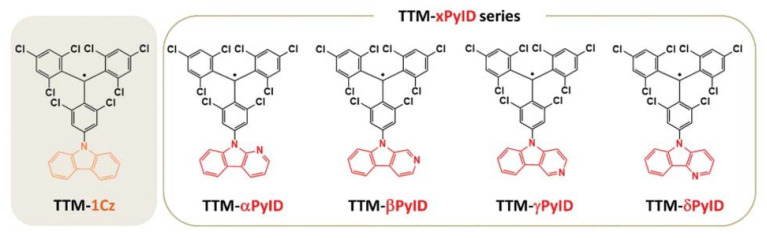
Molecular structures of TTM-based organic radicals modified with carbazoline at different N positions. Reprinted with permission from ref. [120]. Copyright 2021, Copyright Journal of Materials Chemistry C.

**Figure 14 molecules-27-01632-f014:**
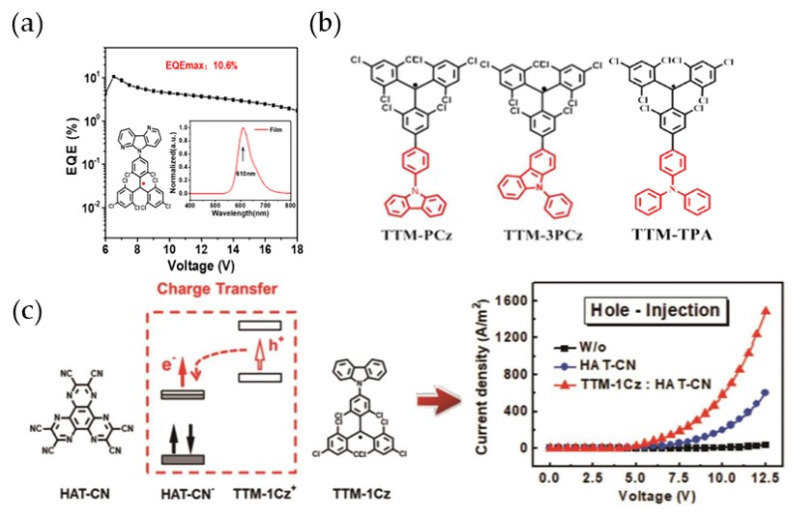
(**a**) TTM-DACz fabricated high-efficiency red-light emission OLED devices. Reprinted with permission from ref. [121]. Copyright 2018, Copyright The Journal of Physical Chemistry Letters. (**b**) The structures of TTM-PCz, TTM-3PCz, and TTM-TPA. Reprinted with permission from ref. [88]. Copyright 2018, Copyright Physical Chemistry Chemical Physics. (**c**) OLED-conductive films prepared by doping 10% TTM-1Cz into HTA-CN. Reprinted with permission from ref. [122]. Copyright 2018, Copyright ACS Applied Materials & Interfaces.

**Figure 15 molecules-27-01632-f015:**
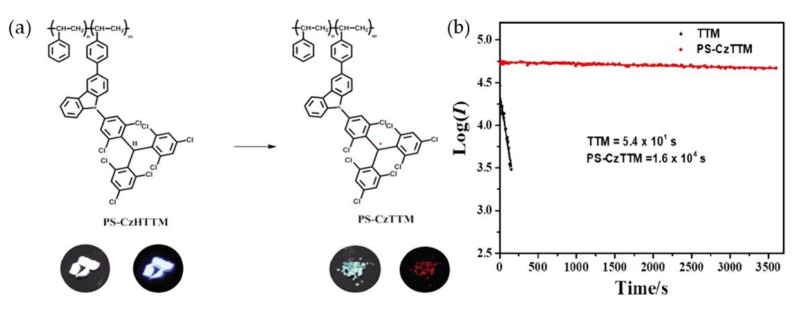
(**a**) Solid-state luminescent radical polymer PS-CzTTM. (**b**) Half-life of TTM and PS-CzTTM. Reprinted with permission from ref. [123]. Copyright 2019, Copyright Materials Horizons.

**Figure 16 molecules-27-01632-f016:**
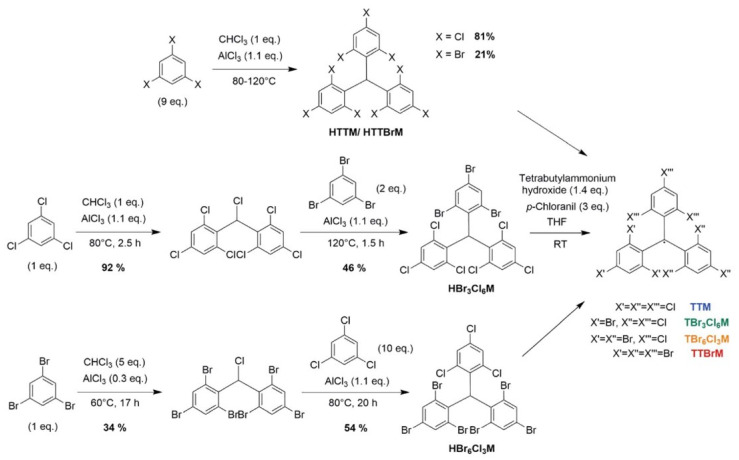
Synthetic route for various halide TTM derivatives. Reprinted with permission from ref. [96]. Copyright 2021, Copyright RSC Advances.

**Figure 17 molecules-27-01632-f017:**
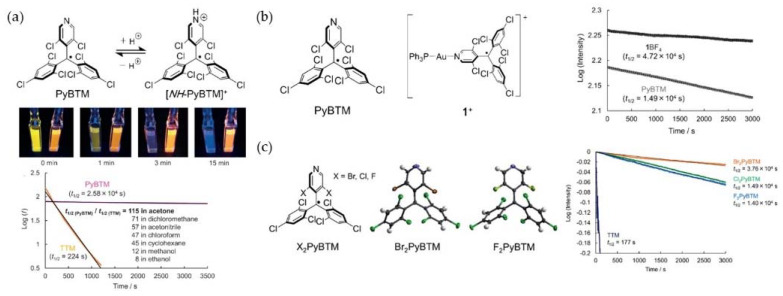
(**a**) Chemical structure, photochemical stability test, and half-life of PyBTM. Reprinted with permission from ref. [98]. Copyright 2014, Copyright Angewandte Chemie International Edition. (**b**) Chemical structure and half-life of metal luminescent radical. Reprinted with permission from ref. [124]. Copyright 2015, Copyright RSC Advances. (**c**) Chemical structure and half-life of X_2_PyBTM (X = F, Cl, Br). Reprinted with permission from ref. [99]. Copyright 2015, Copyright RSC Advances.

**Figure 18 molecules-27-01632-f018:**
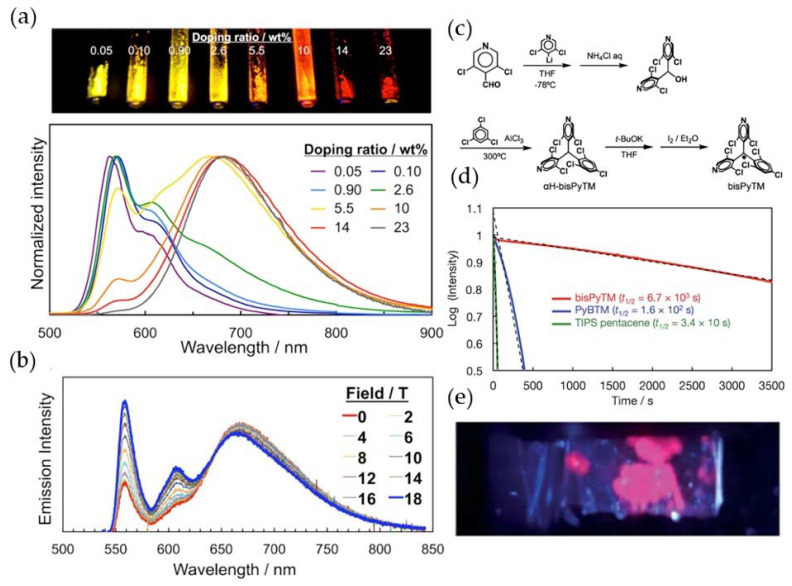
(**a**) Emission spectra of H-PyBTM crystals doped with different concentrations of PyBTM (λex = 370 nm) and their photographs (λex = 365 nm). (**b**) 10% PyBTM doped into H-PyBTM crystals at 4.2 K with different absorption spectra under magnetic field strength. Reprinted with permission from ref. [100]. Copyright 2018, Copyright Angewandte Chemie International Edition. (**c**) Synthetic route of bisPyTM. (**d**) Emission decay curves at 355 nm (bisPyTM), 370 nm (PyBTM), and 310 nm (TIPS pentacene) in dichloromethane. (**e**) Photograph of bisPyTM crystals at 77 K under UV light at 365 nm. Reprinted with permission from ref. [101]. Copyright 2018, Copyright Chemical Science.

**Figure 19 molecules-27-01632-f019:**
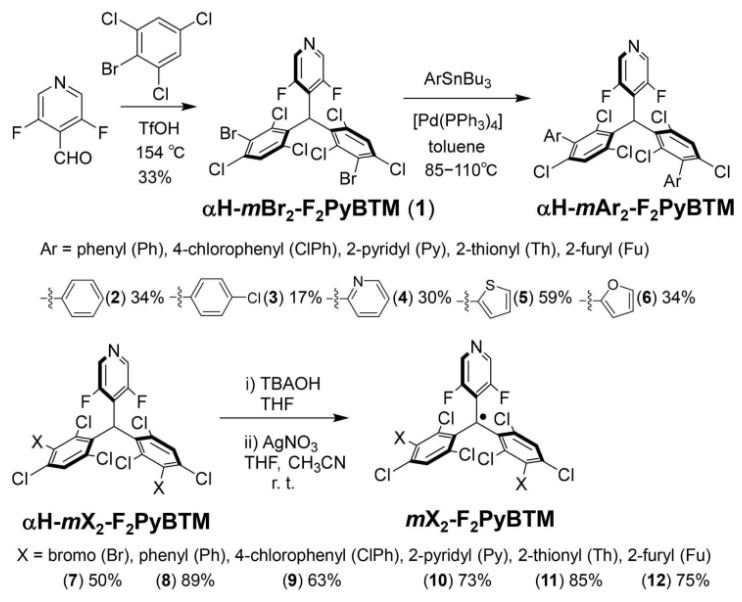
Synthesis of PyBTM-based organic radicals with different halogen atoms and electron-donating-capable substituents. Reprinted with permission from ref. [102]. Copyright 2021, Copyright Chemistry—An Asian Journal.

**Figure 20 molecules-27-01632-f020:**
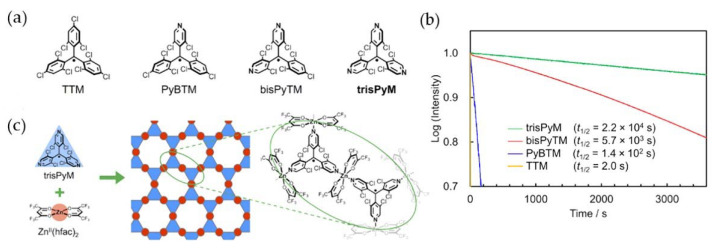
(**a**) Molecular structure and (**b**) half-life of TTM, PyBTM, bisPyTM, and trisPyM. (**c**) Molecular structure of trisZn. Reprinted with permission from ref. [103]. Copyright 2021, Copyright Journal of the American Chemical Society.

**Figure 21 molecules-27-01632-f021:**
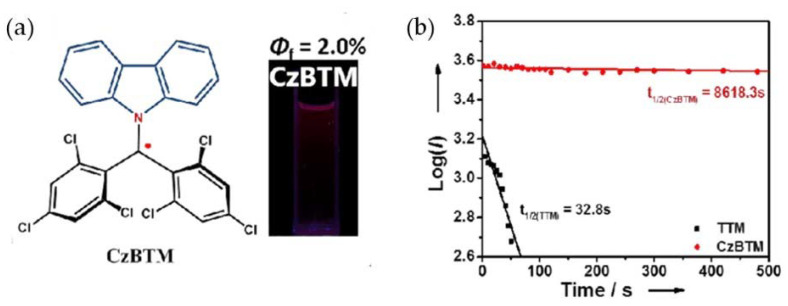
(**a**) Molecular structure and (**b**) half-life of CzBTM. Reprinted with permission from ref. [105]. Copyright 2018, Copyright Angewandte Chemie International Edition.

**Figure 22 molecules-27-01632-f022:**
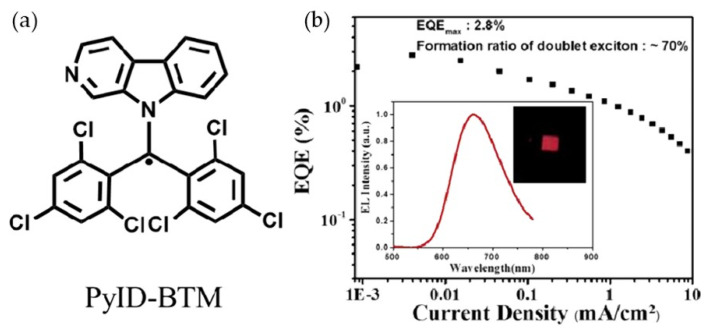
(**a**) Molecular structures of PyID-BTM. (**b**) EQE of fabricated OLED devices. Reprinted with permission from ref. [106]. Copyright 2018, Copyright Journal of Materials Chemistry C.

**Table 1 molecules-27-01632-t001:** PLQE of different triarylmethyl radicals.

Molecule	Solvent	Λf ^a^ (nm)	Φ_f_	Ref.
PTM-PCz	cyclohexane	673	0.44	[88]
PTM-3PCz	cyclohexane	679	0.57	[88]
PTM-TPA	cyclohexane	767	0.26	[88]
PTM-3PCz	cyclohexane	663	0.043	[88]
PTM-TPA	cyclohexane	664	0.29	[88]
3I-PTM^R b^	cyclohexane	605	0.016	[89]
TTM	cyclohexane	680	0.54	[89]
TTM-1Cz	/	611	0.91	[90]
TTM-2Bi	cyclohexane	562	0.008	[91]
TTM-3Bi	cyclohexane	692	0.58	[92]
TTM-αPyID	chloroform	588	0.16	[93]
TTM-βPyID	chloroform	593	0.30	[93]
TTM-γPyID	cyclohexane	599	0.63	[94]
TTM-δPyID	cyclohexane	610	0.98	[94]
2αPyID-TTM	cyclohexane	598	0.37	[94]
2δPyID-TTM	cyclohexane	614	0.89	[94]
TTM-PCz	cyclohexane	608	0.65	[95]
TTM-3PCz	cyclohexane	627	0.92	[95]
TBr3Cl6M	DCM	/	0.018	[96]
TBr6Cl3M	DCM	/	0.012	[96]
TTBrM	DCM	/	0.008	[96]
TTM-DPA	cyclohexane	705	0.65	[97]
TTM-DBPA	cyclohexane	748	0.28	[97]
TTM-DFA	cyclohexane	809	0.05	[97]
PyBTM	EPA ^c^ (77 K)	/	0.81	[98]
Br2PyBTM	chloroform	593	0.02	[99]
F2PyBTM	chloroform	566	0.06	[99]
PyBTM ^d^	/	680	0.89	[100]
bisPyTM	dichloromethane	650	0.009	[101]
mBr2-F2PyBTM	EPA ^c^ (77 K)	/	0.11	[102]
mPh2-F2PyBTM	EPA ^c^ (77 K)	/	0.12	[102]
mClPh2-F2PyBTM	EPA ^c^ (77 K)	/	0.11	[102]
mPy2-F2PyBTM	EPA ^c^ (77 K)	/	0.07	[102]
trisPyM	CH2Cl2	700	0.0085	[103]
metaPyBTM	dichloromethane	571	0.017	[104]
CzBTM	cyclohexane	697	0.020	[105]
PyID-BTM	cyclohexane	664	0.195	[106]

^a^ Wavelength of the maximum fluorescence emission. ^b^ 4% of 3I-PTM^R^ radically doped 3I-PTM^H^ solids. ^c^ Diethyl ether: isopentaane: ethanol at 5:5:2 *v*/*v* mixture cooled with liquid nitrogen. ^d^ 5% of PyBTM radically doped αH-PyBTM solids.

## Data Availability

Not applicable.

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
