# Peer review of "Research Progress on Triarylmethyl Radical-Based High-Efficiency OLED"

_molecules, 2022, doi:10.3390/molecules27051632_

Round 1

Reviewer 1 Report

The authors present the review concerning the application of organic radicals in the construction of highly efficient OLEDs. The aim of the review is clearly stated, the useful introduction to the context of OLEDs is provided, the discussion on the research progress in the selected sub-field is well presented. Thus, I recommend the publication of this paper in the Molecules journal after taking into account two following points:

1. The introduction should contain a paragraph regarding the other (from the application of radicals) research trends in the quest for novel materials applicable in OLEDs, e.g. on metal-based luminophores, including chiral luminophores for CP-OLEDs. One sentence could be given on how the idea of CP-OLEDs might be realized using the radical systems.

2. The short table summarizing the best-performance radical systems in the context of their OLED applications should be provided.

Author Response

Response to comments of Referee 1.

We would like to thank Referee 1 for his/her valuable comments. 

Reviewer #1: The authors present the review concerning the application of organic radicals in the construction of highly efficient OLEDs. The aim of the review is clearly stated, the useful introduction to the context of OLEDs is provided, the discussion on the research progress in the selected sub-field is well presented. Thus, I recommend the publication of this paper in the Molecules journal after taking into account two following points.

1. The introduction should contain a paragraph regarding the other (from the application of radicals) research trends in the quest for novel materials applicable in OLEDs, e.g. on metal-based luminophores, including chiral luminophores for CP-OLEDs. One sentence could be given on how the idea of CP-OLEDs might be realized using the radical systems.

Reply: Thanks for the thoughtful advice. Added as suggested in introduction, and discussed in Page 2, Line 48-58.

2. The short table summarizing the best-performance radical systems in the context of their OLED applications should be provided.

Reply: Added as suggested in Table 1, and discussed in Page 5, Line 159-161.

Reviewer 2 Report

It is always a difficult task/concern to judge extended reviews. The type of radicals considered here are all triarylmethyl based radicals. Only on page 4 is mentioned ‘….more stable organic radicals have been discovered ref 81-85’, however, that are strange citations. For sure stable radicals have been searched for 100 years and more general citations could be made like Hicks Stable radicals Wiley 2010 or a recent review on Air-stable organic radicals Adv. Mater 2020, 32, 1908015. But they have often low or no fluorescence and sure mainly triarylmethyl radicals were used so far for OLEDs. To state on p2 ‘we summarize recent progress of different organic radicals… ‘ is thus misleading. No verdazyl or nitronylnitroxides or any other group of radicals are described, only triarylmethylradicals.

This may also be considered for a tile change which appears too broad instead of Organic Radicals I would thus suggest Triarylmethyl Radicals in the Title.  

The classification of OLED is taken from ref 59 but concerning Fig.  2b and 2c they are wrongly assigned in Figure text.

The title of Figure 3 is misleading. No mechanism is presented just the frontier orbitals of the excited state spin configurations for closed shell and doublet open shell molecules are given.

Over all this review seems timely and could be considered for publication after revision.

Author Response

Response to comments of Referee 2

We would like to thank Referee 2 for his/her valuable comments.

Reviewer #2: Over all this review seems timely and could be considered for publication after revision.

1. It is always a difficult task/concern to judge extended reviews. The type of radicals considered here are all triarylmethyl based radicals. Only on page 4 is mentioned ‘….more stable organic radicals have been discovered ref 81-85’, however, that are strange citations. For sure stable radicals have been searched for 100 years and more general citations could be made like Hicks Stable radicals Wiley 2010 or a recent review on Air-stable organic radicals Adv. Mater 2020, 32, 1908015. But they have often low or no fluorescence and sure mainly triarylmethyl radicals were used so far for OLEDs. To state on p2 ‘we summarize recent progress of different organic radicals… ‘ is thus misleading. No verdazyl or nitronylnitroxides or any other group of radicals are described, only triarylmethylradicals.

Reply: Revised as suggested in main text. We have replaced the strange reference with a more appropriate one (Air-stable organic radicals Adv. Mater 2020, 32, 1908015.). Also, we've changed the misleading language. Thanks for your valuable suggestion.

2. This may also be considered for a tile change which appears too broad instead of Organic Radicals I would thus suggest Triarylmethyl Radicals in the Title.

Reply: Revised as suggested in Title and main text. “Triarylmethyl Radicals” would be more precise to show the main idea of this paper.

3. The classification of OLED is taken from ref 59 but concerning Fig. 2b and 2c they are wrongly assigned in Figure text.

Reply: Revised as suggested. The manuscript was further carefully checked to avoid these mistakes.

4. The title of Figure 3 is misleading. No mechanism is presented just the frontier orbitals of the excited state spin configurations for closed shell and doublet open shell molecules are given.

Reply: Revised as suggested in Title of Figure 3 in Page 4, Line 139.